# Phosphorylation of Syntaxin 4 by the Insulin Receptor Drives Exocytic SNARE Complex Formation to Deliver GLUT4 to the Cell Surface

**DOI:** 10.3390/biom13121738

**Published:** 2023-12-02

**Authors:** Dimitrios Kioumourtzoglou, Hannah L. Black, Mohammed Al Tobi, Rachel Livingstone, John R. Petrie, James G. Boyle, Gwyn W. Gould, Nia J. Bryant

**Affiliations:** 1Department of Biology, University of York, Heslington YO10 5DD, UK; dimi.kioumourtzoglou@york.ac.uk (D.K.);; 2Institute of Molecular Cell and Systems Biology, College of Medical Veterinary and Life Sciences, University of Glasgow, Glasgow G12 8QQ, UK; toubim@squ.edu.om (M.A.T.);; 3Institute of Cardiovascular and Medical Sciences, College of Medical Veterinary and Life Sciences, University of Glasgow, Glasgow G12 8QQ, UK; 4School of Medicine, Dentistry & Nursing, College of Medical Veterinary and Life Sciences, University of Glasgow, Glasgow G12 8QQ, UK; james.boyle@glasgow.ac.uk; 5Strathclyde Institute of Pharmacy and Biomedical Sciences, University of Strathclyde, 161 Cathedral Street, Glasgow G4 ORE, UK

**Keywords:** membrane traffic, Syntaxin, SNARE complex: insulin action, GLUT4

## Abstract

A major consequence of insulin binding its receptor on fat and muscle cells is the stimulation of glucose transport into these tissues. This is achieved through an increase in the exocytic trafficking rate of the facilitative glucose transporter GLUT4 from intracellular stores to the cell surface. Delivery of GLUT4 to the cell surface requires the formation of functional SNARE complexes containing Syntaxin 4, SNAP23, and VAMP2. Insulin stimulates the formation of these complexes and concomitantly causes phosphorylation of Syntaxin 4. Here, we use a combination of biochemistry and cell biological approaches to provide a mechanistic link between these observations. We present data to support the hypothesis that Tyr-115 and Tyr-251 of Syntaxin 4 are direct substrates of activated insulin receptors, and that these residues modulate the protein’s conformation and thus regulate the rate at which Syntaxin 4 forms SNARE complexes that deliver GLUT4 to the cell surface. This report provides molecular details on how the cell regulates SNARE-mediated membrane traffic in response to an external stimulus.

## 1. Introduction

Insulin is a critical regulator of whole-body glucose homeostasis. A major action of insulin is to lower plasma glucose by stimulating uptake into muscle and adipose tissue. Insulin-binding cell surface receptors initiate signalling events that culminate in the translocation of the facilitative glucose transporter GLUT4 from internal stores to the plasma membrane [1]. Insulin-stimulated translocation of GLUT4 to the cell surface is a regulated membrane trafficking event that, like all membrane trafficking events, is mediated by SNARE proteins, a highly conserved family found in all eukaryotes [2]. SNARE proteins on donor vesicles (v-SNAREs) form a tightly bound complex with specific SNARE proteins in the appropriate target membrane (t-SNAREs) orchestrating the docking of the membranes and facilitating membrane fusion by providing the necessary energy for the fusion of the two lipid bilayers [3]. Delivery of GLUT4 to the cell surface from internal stores is mediated in part by the formation of SNARE complexes between the t-SNAREs Syntaxin 4 (Sx4) and SNAP23 on the plasma membrane, and the v-SNARE VAMP2 on internal insulin-responsive vesicles (IRVs) [1,4,5].

SNARE complex assembly is subject to multiple layers of regulation, enabling the cell to regulate membrane traffic. One way in which cells regulate SNARE complex assembly is through a conformational switch in the syntaxin (Qa) SNARE [6]. Syntaxins are a sub-family of SNARE proteins that, in addition to the defining coiled-coil SNARE motif that participates in the SNARE complex along with three others from the partner SNAREs, contain an autonomously folded N-terminal domain known as the H_abc_ domain [6]. Several syntaxins, including Sx4, have been shown to adopt two distinct conformations: open and closed [6,7,8,9,10,11]. In the closed conformation, the H_abc_ domain forms intramolecular contacts with the SNARE motif, rendering it inaccessible for binding to partner SNARE proteins, whereas, in an open conformation, the SNARE motif is freed from this autoinhibition and is available for SNARE complex formation [6,9,10,11]. Given that SNARE complex formation is sufficient to drive bilayer fusion, regulation of this conformational switch as a way to regulate SNARE complex assembly affords the cell a mechanism to regulate membrane traffic [6]. Post-translational modification of Syntaxin offers one such potential regulatory mechanism [10,12,13].

We have previously shown that insulin stimulates the formation of Sx4/SNAP23/VAMP2 complexes in 3T3-L1 adipocytes, identifying this process as a mechanism by which insulin increases the exocytosis of GLUT4 [14]. Insulin exerts its effects through binding cell surface receptors, activating their tyrosine kinase activity [15]. A quantitative mass spectrometry approach to identify insulin-induced tyrosine phosphorylation sites in 3T3-L1 adipocytes identified two residues in Sx4 (Y115 and Y251) as showing a ≥3-fold increase in phosphorylation in response to insulin [16]. The consequences of this phosphorylation are as yet unknown; however, it may represent an important step in the communication between signalling and trafficking pathways of insulin action especially since phosphorylation of several SNARE proteins has been demonstrated to alter interaction with their SNARE binding partners [11,13,17,18,19,20].

In this study, we examined the effects of phosphorylation of Y115 and Y251 of Sx4. We demonstrate that these residues are directly phosphorylated by the kinase domain of the insulin receptor and provide evidence in support of a hypothesis whereby insulin-stimulated phosphorylation of Sx4 on Y115 and Y251 triggers a conformational switch to drive the formation of Sx4/SNAP23/VAMP2 SNARE complexes, resulting in the delivery of GLUT4 to the cell surface.

## 2. Materials and Methods

### 2.1. Plasmids

Bacterial expression vectors encoding GST, WT-Sx4-GST (C-terminally GST-tagged Sx4 cytosolic domain), His-SNAP23 (N-terminally His-tagged SNAP23), and Open-Sx4-GST (C-terminally GST-tagged Sx4 cytosolic domain harbouring two mutations in the hinge region [L173A/E174A]) have all been described elsewhere [7,14]. Y115-Sx4-GST (C-terminally GST-tagged Sx4 cytosolic domain harbouring a Y115E mutation), Y251E-Sx4-GST (C-terminally GST-tagged Sx4 cytosolic domain harbouring a Y251E mutation), and Phos-Sx4-GST (C-terminally GST-tagged Sx4 cytosolic domain harbouring two mutations Y115E and Y251E) were generated from the plasmid encoding WT-Sx4-GST [7] by site-directed mutagenesis and were all confirmed by DNA sequencing. The mammalian expression vectors WT-Sx4-Myc and Phos-Sx4-Myc (encoding full-length Sx4 carrying a double Myc-tag positioned between residues 36 and 37, i.e., between the N-terminus and H_abc_ domain, and that with Y115 and Y251 both mutated to glutamic acid residues) were constructed in pCR3.1 (Invitrogen, Waltham, MA, USA) using the custom gene synthesis service by GenScript, Piscataway, NJ, USA.

### 2.2. Antibodies

Commercial sources of antibodies were as follows: Rat monoclonal clone 3F10 anti-HA (Roche applied sciences, Penzberg, Germany, 11867423001), mouse monoclonal anti-His (Sigma-Aldrich, Dorset, UK, H1029), mouse monoclonal anti-VAMP2 (Synaptic Systems, Göttingen, Germany, 104211), rabbit polyclonal anti-Sx4 (Synaptic Systems, Göttingen, Germany, 110042), mouse monoclonal anti-phosphotyrosine (BD Biosciences, PY20, 610000, San Jose, CA, USA), mouse monoclonal clone 9E10 anti-Myc (Sigma-Aldrich, Dorset, UK, M4439), mouse monoclonal anti-Myc (Cell Signaling, 2276), rabbit polyclonal anti-SNAP23 (Synaptic Systems, Göttingen, Germany, 111203 BT), rabbit polyclonal anti-VAMP2 (Cell Signaling, 13,508), rabbit monoclonal anti-MUNC18c (Abcam, Cambridge, UK, ab175238), anti-mouse IgG(whole molecule) peroxidase conjugate produced in goat (Sigma-Aldrich, Dorset, UK, A4416), Anti-Rabbit IgG (whole molecule) peroxidase conjugate antibody produced in goat (Sigma-Aldrich, Dorset, UK, A0545), anti-rat IgG (whole molecule) cross-adsorbed secondary antibody, Alexa Fluor™ 647 produced in goat (ThermoFisher Scientific, A-21247), anti-mouse IgG (whole molecule) DyLight™ 405 produced in goat (Jackson Immunoresearch, West Grove, PA, USA, 115-475-003).

### 2.3. Expression and Purification of Recombinant Proteins

Affinity chromatography was used to purify GST-fusion and His-tagged proteins produced in *E. coli*, as previously described [7].

### 2.4. In Vitro Phosphorylation of Sx4 by CIRK

Recombinant CIRK, provided by Professor Gustav E. Lienhard (Dartmouth Medical School), was activated by autophosphorylation as previously described [21]. CIRK and Syntaxin 4 (0.3 μM and 0.72 μM final concentrations, respectively) were mixed in filter-sterilised reaction buffer (50 mM HEPES pH 7.5, 4 mM MnCl_2_, 0.2 mM DTT, and 100 µM ATP Na-salt). The assay was carried out at 30 °C for 150 min. For MS analysis, bands corresponding to Sx4 were excised from the gel and analysed using in-gel trypsin digestion and nano-LC MS/MS at the FingerPrints Proteomic Facility, University of Dundee.

### 2.5. In Vitro SNARE Complex Assembly Assay

Equimolar concentrations, ~5 µM, of WT-Sx-4-GST (or mutant versions thereof), His-SNAP23, and VAMP2-GST (or GST controls) were combined in a total volume of 1000 μL in reaction buffer (25 mM HEPES, 0.4 M KCl, 10% (*w*/*v*) glycerol, pH 7.4) plus 100 μg/mL BSA (Tocris, Bristol, UK) and incubated at 4 °C, with rotation, for 20 min. Samples were immediately analysed by SDS-PAGE and immunoblotting.

### 2.6. Limited Proteolysis

A total of 1.2 nM chymotrypsin (Sigma-Aldrich) was incubated with 5 µM WT-Sx4-GST or a mutant version thereof (final concentrations) in PBS at room temperature. The reaction was terminated by the addition of SDS-PAGE sample buffer (100 mM Tris, HCl pH 6.8, 4% (*w*/*v*) SDS, 20% (*v*/*v*) glycerol, 0.2% (*w*/*v*) bromophenol blue, and 10% (*v*/*v*) β-mercaptoethanol), and heating to 95 °C for 5 min.

### 2.7. In Vitro Binding Assay

A total of 1–5 μg WT-Sx4-GST or a mutant version thereof was immobilised to 60 μL of glutathione Sepharose beads by incubation for 3 h at 4 °C with rotation followed by washes with PBS. Subsequently, the beads carrying the protein were incubated with an excess of either His-SNAP23 or thrombin-cleaved VAMP2 in 500 μL binding buffer (PBS, 100 μ/mL BSA) at 4 °C with rotation for the indicated times. Following incubation, beads were collected by centrifugation, washed, and resuspended in 50 μL of SDS-PAGE sample buffer heated at 95 °C for 5 min. Samples were then analysed by SDS-PAGE and immunoblotting.

### 2.8. Immunoblotting and Coomassie Staining

Protein samples, separated by SDS-PAGE (10% acrylamide). The gel was either immersed in Coomassie staining solution (0.05% (*w*/*v*) Coomassie brilliant blue R250, 50% (*v*/*v*) methanol, and 10% (*v*/*v*) acetic acid) for at least 45 min, followed by destaining with 15% (*v*/*v*) methanol, and 15% (*v*/*v*) acetic acid, or it was transferred to nitrocellulose membrane (0.2 µm, Bio-Rad) for immunoblot analysis. The membrane was labelled with primary antibodies (1–5 μg/mL), washed with TBST, and subsequently, labelled with species-specific HRP-conjugated secondary antibodies (Thermo Fisher Scientific, Horsham and Loughborough, UK, 1 μg/mL). Peroxidase activity was detected using Western Lightning Chemiluminescence Reagent (GE Healthcare). The molecular migration position of transferred proteins was compared with the Precision Plus All Blue Protein Ladder 10–250 kD (Thermo Fisher Scientific, Horsham and Loughborough, UK). Densitometry analysis of immunoblots and Commassie-stained gels was carried out using ImageJ software (version 1.50 g).

### 2.9. Isothermal Titration Calorimetry

Isothermal titration calorimetry was performed using a MicroCal VP-ITC machine (Malvern Scientific Instruments, Malvern, UK). Syntaxins (around 300 µM) were titrated with increasing amounts of SNAP23 (64 µM), and the raw data were fitted using a nonlinear least squares routine using Microcal Origin software (version 7.0).

### 2.10. Cell Culture

All cell lines were maintained at 37 °C in a 5% CO2 atmosphere. The HeLa cell line stably expressing HA-GLUT4-GFP had been previously generated [22], and was cultured in Dulbecco’s Modified Eagle’s Medium (DMEM) (Gibco), supplemented with 10% (*v*/*v*) Foetal Calf Serum (Gibco, Paisley, UK), 1% (*v*/*v*) Glutamax (ThermoFisher Scientific, Horsham and Loughborough, UK) and 1% (*v*/*v*) Penicillin-Streptomycin (10,000 U/mL; Gibco). 3T3L-1 mouse embryo fibroblasts cell lines, stably expressing WT-sx4-Myc or Phos-Sx4-Myc were generated by transfecting 3T3-L1 cells with the mammalian vectors encoding the corresponding constructs using GeneCellin (BioCellChallenge, Paris, France, following the manufacturer’s instructions. Transfectants were selected in growth medium (DMEM 10% (*v*/*v*) New-born Calf Serum (Gibco, Paisley, UK), 1% Glutamax (*v*/*v*), and 1% Penicillin-Streptomycin) supplemented with 700 µg/mL G418 (Gibco) and then maintained in growth medium with 700 µg/mL G418 until single colonies appeared which subsequently were isolated and expanded. Successful clones were confirmed by immunoblotting. Briefly, cells washed with PBS three times and then scraped in adipocyte lysis buffer (150 mM NaCl, 20 mM Tris pH8, 2 mM EDTA, 0.5% (*v*/*v*) Triton x-100) 50 µL per well-6 well plate, and homogenized by pulling 10 times through a 24 gauge, and subsequently two times through a 26-gauge needle. The insoluble materials were pelleted by centrifugation for 20 min at 15,000× *g* at 4 °C and the clear lysate was prepared for SDS-PAGE analysis by the addition of SDS-PAGE sample buffer heated at 95 °C for 5 min.

### 2.11. HeLa Cells Surface Immunostaining and Flow Cytometry

Cells were seeded on coverslips in 6 well plates and transfected using GeneCellin (BioCellChallenge, Paris, France), following the manufacturer’s instructions, 48 h prior to analysis. Cells were serum starved for two hours in serum-free DMEM to set cell surface GLUT4 to basal levels. Cells were washed thoroughly with cold PBS before being fixed with 4% (*w*/*v*) methanol-free PFA for 20 min followed by additional washes with cold PBS before blocking with 2% (*w*/*v*) BSA in PBS for 45 min. GLUT4 was then surface stained with 50 µg/mL anti-HA 3F10 primary antibody in a blocking solution for 45 min. Following three washes with blocking solution, 4 µg/mL Alexa Fluor 647 (ThermoFisher, Horsham and Loughborough, UK) coupled secondary antibody in blocking solution was then applied for 45 min followed by three washes. To stain for intracellular Sx4-Myc, cells were then fixed again using 4% electron microscopy grade, methanol-free PFA for 20 min to fix anti-HA antibodies in place. Cells were washed three times, and then permeabilised and blocked with PBS plus 0.1% (*w*/*v*) saponin and 2% (*w*/*v*) BSA for 45 min. The cells were then stained for Sx4-Myc expression using 40 µg/mL anti-Myc antibody in a blocking solution for 45 min. Cells were then washed again three times and 4 µg/mL anti-mouse Alexa Fluor 405 antibody was applied, and diluted in a blocking solution for 30 min. Following surface immunostaining as described above, cells were dissociated from the tissue culture dish using a cell lifter. Dissociated cells were collected and centrifuged at 600× *g* for 8 min at 4 °C before resuspension in PBS plus 2% BSA. Samples were analysed on a Beckman Coulter CyAn ADPs flow cytometer. Data were collected and analysed using Summit Software v4.3. A minimum of 10,000 cells were analysed for each experimental condition, performed on at least three biological replicates.

### 2.12. 3T3-L1 Cell Differentiation and Proximity Ligation Assay (PLA)

PLA was performed on mouse embryo fibroblast-derived 3T3-L1 adipocytes differentiated on Labtech 8-chamber slides as described previously [14]. Briefly, cells were washed 3 times with PBS prior to fixation in 3% *w*/*v* methanol-free PFA for 30 min at room temperature. Cells were then washed twice with 20 mM glycine in PBS, blocked, and permeabilized in 0.1% (*w*/*v*) saponin, 2% (*w*/*v*) BSA, and 20 mM glycine in PBS for 30 min. Primary antibody incubations were performed in the same solution overnight at 37 °C in a humidity incubator (50 μg/mL), after which PLA was performed according to the instructions of the manufacturer (Duolink, Merk, Feltham, UK). Signals were visualized using a Carl Zeiss LSM 880 AiryScan fluorescence system. Cell Profiler software (version 4.0.5) was used for PLA signal quantification from 300 to 500 cells per experiment, performed on at least three biological replicates.

### 2.13. Statistical Analyses

Statistical analysis was performed using GraphPad Prism 8. A standard two-way analysis of variance (ANOVA), assuming a normal distribution, was used for multiple comparisons, and a statistical difference was defined as a *p*-value of <0.05. All experiments were performed at least 3 times independently. Data is reported as mean ± SEM, *p*-values, and n numbers for each experiment are defined in the figure legends.

## 3. Results

### 3.1. Sx4 is Directly Phosphorylated by the Insulin Receptor In Vitro

A quantitative mass spectrometry study identified two tyrosine residues in Sx4 that undergo an increase in phosphorylation in response to insulin; Y115 and Y251 [16]. However, this study did not identify the kinase responsible for this phosphorylation. The insulin-signalling cascade is initiated by the activation of insulin receptor tyrosine kinase activity. We, therefore, sought to examine whether Sx4 can be directly tyrosine phosphorylated in vitro by activated insulin receptor kinase. To this end, we used recombinant activated cytosolic insulin receptor kinase (CIRK; corresponding to residues 941–1343 of the β-subunit of the insulin receptor) purified from baculovirus-infected S9 insect cells [23,24]. Using an in vitro phosphorylation assay, we were able to show that Sx4 can be phosphorylated directly by CIRK (Figure 1A) on both Y115 and Y251 (Figure 1B). The observation that a version of Sx4 in which both Y115 and Y251 have been mutated to a non-phosphorylatable residue is not phosphorylated in this assay is consistent with the notion that these residues represent the only two phospho-acceptor sites for activated insulin receptor kinase within the cytosolic domain of Sx4 (Figure 1B). This was also confirmed by MS analysis of recombinant proteins after in vitro phosphorylation using nanoLC MS/MS, which identified Y115 and Y251 as the only phospho-sites.

### 3.2. Phos-Sx4 Increases SNARE Complex Formation In Vitro

Having demonstrated that Sx4 can be directly phosphorylated by CIRK on residues Y115 and Y251 (Figure 1), we sought to investigate the functional consequences of this phosphorylation. We have previously shown that insulin stimulates the assembly of functional Sx4/SNAP23/VAMP2-containing SNARE complexes in 3T3-L1 adipocytes [14] and therefore hypothesised that this may be regulated by insulin-stimulated phosphorylation of Sx4 at Y115 and Y251. To test this, we examined the ability of a version of Sx4’s cytosolic domain mutated to mimic the phosphorylation of Y115 and Y251 to form SNARE complexes with the cytosolic domains of SNAP23 and VAMP2. Phos-Sx4-GST was generated by mutating both tyrosine residues to bulky, negatively charged glutamate residues within a version of Sx4 harbouring a GST-tag in place of its transmembrane domain, a construct previously demonstrated to form SNARE complexes with the cytosolic domains of VAMP2 and SNAP23 [7,14,21]. Figure 2 shows that, compared to the wildtype protein, this version of Sx4-GST shows an increased propensity to form SNARE complexes as assessed by the formation of an SDS-resistant complex of ~150 kd (representing a ternary complex containing the Sx4, SNAP-23, and VAMP2 constructs used in this assay [14]). The increased complex formation observed for Phos-Sx4 in this assay is comparable to that seen for a version of Sx4 (Sx4-open) previously shown to form SNARE complexes at a faster rate than wildtype [14] (Figure 2).

### 3.3. Conformational Changes in Phos-Sx4 Increase Interaction with SNARE Binding Partners

The observation that Phos-Sx4 shows an enhanced ability to form SNARE complexes comparable to that seen with the Sx4-open mutant (Figure 2), led us to hypothesize that phosphorylation of Sx4 at residues 115 and 251 facilitates the regulatory switch of Sx4 from a closed to an open conformation [7,8]; a conformational shift that represents an evolutionarily conserved mechanism allowing eukaryotic cells to regulate SNARE complex formation [6]. The more open conformation of Sx4-open compared to the wildtype protein is reflected by its increased susceptibility to proteolysis in vitro [7]. Figure 3 shows that the Phospho-Sx4 mutant is also more susceptible to degradation in a limited proteolysis assay, consistent with it adopting a more open conformation (Figure 3A,B). In addition to the ternary SNARE complex, Sx4 also forms binary complexes with SNAP23 and VAMP2 both in vivo and in vitro [14,25,26]. Figure 3 also shows that Phospho-Sx4 mirrors the enhanced ability of Sx4-open to form binary complexes with either SNAP23 (Figure 3C,E) or VAMP2 (Figure 3D,F). The data in Figure 3 support the hypothesis that phosphorylation of Sx4 at residues 115 and 251 facilitates a switch of Sx4 to an open conformation bestowing an enhanced ability to bind its SNARE partners. Further, in support of these observations, isothermal titration calorimetry revealed that phospho-Syntaxin 4 exhibited increased affinity for SNAP23 compared to wildtype (K_d_ 1.22 versus 5.6 µM, respectively).

### 3.4. Phos-Sx4 Increases Levels of GLUT4 at the Cell Surface and Enhances SNARE Protein Interactions

To extend our biochemical studies indicating that Phospho-Sx4 enhances SNARE complex formation through a change in protein conformation (Figure 2 and Figure 3) we investigated the effects of Phospho-Sx4 cells. HeLa cells stably expressing a version of GLUT4 harbouring an exofacial HA tag and a C-terminal GFP tag are widely used as a model to study insulin-regulated GLUT4 traffic [27,28,29]. The amount of GLUT4 at the surface of these cells can be examined by immuno-staining with an anti-HA antibody in the absence of any cell permeabilisation to identify GLUT4 at the plasma membrane with GFP signal representing total GLUT4 present in the cell. Cell surface GLUT4 can be accurately quantified using flow cytometry to measure the mean fluorescence intensity (MFI) of surface GLUT4 (HA signal) relative to the MFI of total cellular GLUT4 (GFP signal) [30]; this approach is particularly powerful as it allows quantification of many thousands of cells per condition, rather than the more subjective approach of using microscopy to image only a few cells at a time. Figure 4 shows when transfected with a myc-tagged version of full-length Sx4 harbouring the phosphomimetic mutations (Y115E and Y251E: Phos-Sx4), shown to enhance the ability of Sx4’s cytosolic domain to interact with its SNARE binding partners (Figure 2 and Figure 3), significantly more HA-GLUT4-GFP is present at the surface compared to control cells transfected with a wild-type version of the same Sx4 construct (WT-Sx4). The myc-tag was included to identify cells expressing the constructs in transient expression experiments and was inserted between residues 36 and 37 of Sx4, N-terminal to the autonomously-folded H_abc_ domain, and importantly the two constructs express to comparable levels in the experiments presented (Figure 4A) allowing comparison of surface levels of HA-GLUT4-GFP between the two populations of cells (Figure 4A,B). The tag was placed here rather than at the extreme N-terminus as tagging at that position compromises Sx4’s interactions with its regulatory protein Munc18C [7].

Given that delivery of GLUT4 to the cell surface is regulated by Sx4, the observation that expression of Phos-Sx4 results in more GLUT4 being present at the cell surface in the absence of insulin stimulation is consistent with our hypothesis that phosphorylation of Sx4 on tyrosine residues 115 and 251 drives its entry into functional SNARE complexes. We have previously used proximity ligation assay (PLA) to demonstrate that insulin drives Sx4-containing SNARE complex formation [14]. To ask whether this is facilitated via phosphorylation of Sx4 we used 3T3-L1 adipocytes stably expressing myc-tagged wildtype or phosphomimetic versions of Sx4 and performed PLA to compare their pairwise associations with their SNARE binding partners SNAP23, VAMP2 and Munc18c (Figure 5). Immunoblotting using anti-myc antibodies indicated that the tagged proteins were expressed at broadly similar levels across the populations of cells used (Figure 5A). Anti-myc antibodies were also used for PLA in conjunction with anti-Sx4 antibodies to quantify expression levels of the Sx4 variants in each individual cell confirming that myc-tagged wild-type Sx4 and myc-tagged phosphomimetic Sx4 are equally expressed in these stable cell lines (Figure 5A,B). We then used the myc-tag to quantify interactions between the over-expressed wild-type and phosphomimetic proteins, excluding any associations with the endogenous Sx4. Figure 5C demonstrates that the introduction of phosphomimetic mutations into Sx4 enhances interactions with all three SNARE binding partners, SNAP23, VAMP2, and Munc18c. It is possible that these data reflect concomitant increases of each binary interaction between Phos-Sx4 and its SNARE partners individually, but given the data in Figure 4 showing that the Phos-Sx4 also increases translocation of GLUT4 to the cell surface—an event that requires SNARE complex formation—we favour the hypothesis that Phos-Sx4 forms SNARE complexes more readily than its wildtype counterpart.

## 4. Discussion

Understanding how SNARE complex formation is controlled in space and time is important, particularly for trafficking systems regulated by extracellular signals [31,32]. This is exemplified by the ability of insulin to promote regulated exocytosis of GLUT4-containing vesicles to the surface of fat and muscle cells via a process involving the t-SNARE comprised of Sx4 and SNAP23, and a v-SNARE (VAMP2) [4]. Previous work from us and others has identified phosphorylation of the Sec1/Munc18 protein, Munc18c, as a key target of insulin action. Phosphorylation of Munc18c by the insulin receptor is thought to rapidly activate the assembly of functional SNARE complexes at the surface of adipocytes [7,33]. A study of the patterns of tyrosine phosphorylation in response to insulin treatment of adipocytes also identified Sx4 as being rapidly phosphorylated in response to insulin [16]. Here, we have dissected the role of this phosphorylation, and provide data to support the hypothesis that insulin-stimulated phosphorylation of Sx4 promotes the assembly of functional Sx4/SNAP23/VAMP2 SNARE complexes.

Sx4 is phosphorylated on Tyr-115 and Tyr-251 in response to insulin both in vivo [16] and in vitro using recombinant cytosolic insulin receptor kinase (Figure 1). Our data using anti-phosphotyrosine antibodies and MS/MS analysis indicates that these are the only residues phosphorylated in vitro. We have previously shown that when combined in vitro, recombinant Sx4, SNAP23 and VAMP2 assemble into an SDS-resistant SNARE complex, and that the rate of assembly of this complex is enhanced when a so-called ‘open’ mutant of Syntaxin 4 is substituted for wildtype Sx4 [7]. We recapitulated this observation here and go on to demonstrate that a phospho-mimetic mutant of Sx4 induces a similar increase in SNARE complex formation (Figure 2). These data are consistent with the notion that phosphorylation of Sx4 by the insulin receptor results in a conformational change that enhances interactions of Sx4 with its binding partners and thus facilitates SNARE complex assembly. This is further supported by quantification of individual binding events, which reveal that phospho-Sx4 exhibits increased kinetics of binding to both SNAP23 and VAMP2 (Figure 3E,F) and that Phos-Sx4 binds SNAP23 with increased affinity compared to Sx4 (Figure 3G).

To build on these studies in a more physiological setting we tested the prediction made from our in vitro data that expression of phospho-Sx4 would enhance delivery of GLUT4 to the surface of cells in the absence of insulin. To allow the quantification of many thousands of cells to test this hypothesis with rigour, we used HeLa cells expressing HA-GLUT4-GFP transiently transfected with a myc-tagged version of either Sx4 or Phos-Sx4. Expression of Phos-Sx4 consistently elevated cell surface levels of GLUT4 in the absence of insulin, consistent with the idea that phosphorylation of Sx4 has resulted in enhanced SNARE complex formation.

Our attempts to study phospho-mimetic and/or phospho-resistant Sx4 mutants in 3T3-L1 adipocytes have been hampered by two issues. Firstly, the presence of high levels of endogenous Sx4 rendered data difficult to interpret, as effects varied likely as a consequence of varying levels of expression of the mutants relative to the endogenous protein (our unpublished data). We also sought to express mutant versions of Sx4 in Sx4 knockout clones, but these attempts resulted in differentiation failure, likely a result of the number of passages (after the original CRISPR protocols) required to generate stable cell lines [34]. We were, however, able to stably express the myc-tagged versions of Syntaxin 4 in wildtype 3T3-L1 adipocytes and used proximity ligation assays to examine pairwise interactions between the SNARE complex members and over-expressed wildtype or phospho-mimetic Sx4 (with the myc-tag allowing us to distinguish the over-expressed versions from the endogenous Sx4). When expressed at comparable levels, phos-Sx4 displays enhanced with all other components of the SNARE complex, SNAP23, VAMP2, and Munc18c (Figure 5). This observation, taken in conjunction with the data presented in Figure 4 showing enhanced cell surface GLUT4 levels in cells over-expressing phos-Syntaxin 4 supports a model in which phosphorylation of Sx4 on Tyr-115/251 enhances the formation of a functional SNARE complex. Tyr-251 is located N-terminal of the SNARE domain, and Tyr-115 is located towards the C-terminus of the Habc domain, near the so-called hinge-region that allows transition between the closed-to-open conformers of Syntaxin 4. AlphaFold predicts that these residues are in close proximity to each other, making it tempting to suggest that their phosphorylation in response to insulin would facilitate the closed-to-open transition thus favoring the conformation of productive SNARE complexes [6,9,10,11]. Our data are consistent with this model.

This study paves the way for investigations into how other regulated trafficking pathways integrate signaling pathways and membrane trafficking machinery and could also hold relevance for the identification of potential pathophysiological mechanisms underpinning disease states including insulin resistance and Type2 Diabetes.

## Figures and Tables

**Figure 1 biomolecules-13-01738-f001:**
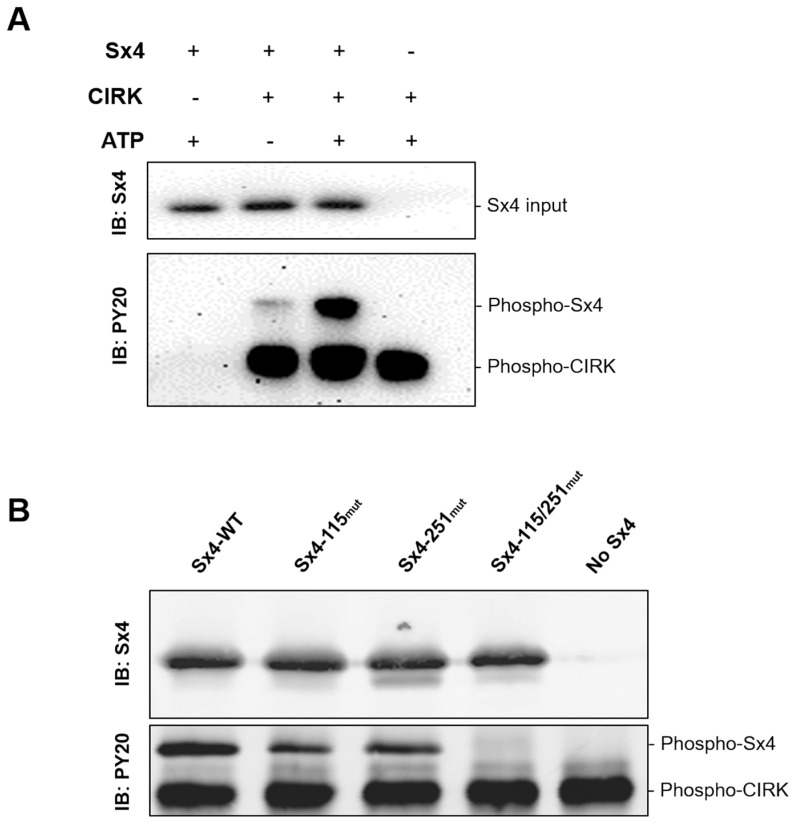
Syntaxin 4 is directly phosphorylated on Y115 and Y251 by activated insulin receptor kinase in vitro. (**A**) 2 µg Sx4-GST was incubated with (+) or without (−) 0.7 µg recombinant activated cytosolic insulin receptor kinase (CIRK) and/or 100 µM ATP at 30 °C for 150 min. Following separation on a 10% SDS-PAGE gel, samples were analysed by immunoblotting for Sx4 (top panel) to visualise Sx4 input, or phospho-tyrosine (PY20, bottom panel): autophosphorylated CIRK and phosphorylated Sx4 are indicated. (**B**) 2 µg Sx4-GST (WT) or a mutant version with either, or both, tyrosine residues 115 and 251 mutated, were incubated with CIRK and ATP and analysed as in (**A**). Representative immunoblots from *n* = 3. Original Western Blotting Figures can be found in Appendix A.

**Figure 2 biomolecules-13-01738-f002:**
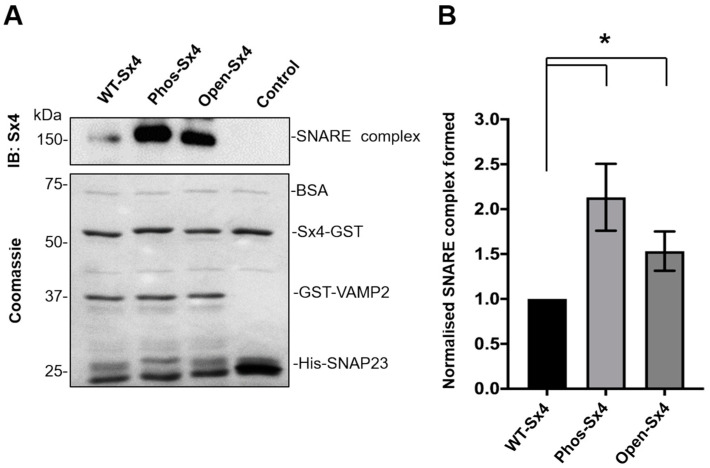
Phosphomimetic mutation of Sx4 increases SNARE complex formation in vitro. A total of 5µM wildtype Sx4-GST (Sx4-WT), or a mutant version, thereof, was incubated with equimolar amounts of GST-VAMP2 and His-SNAP23 at 4 °C for 20 min. (**A**) SDS-PAGE followed by immunoblot analysis was used to assess the ability of the GST-Sx4 variants to form SNARE complexes with GST-VAMP2 and His-SNAP23. Coomassie staining was used to visualise the input proteins for each assay. (**B**) ImageJ software was used to compare the amount of the GST-Sx4 variants assembled into complexes with GST-VAMP2 and His-SNAP23 (normalized to WT). Error bars represent  ± SEM, from 3 experiments * *p* < 0.05. Original Western Blotting Figures can be found in Appendix A.

**Figure 3 biomolecules-13-01738-f003:**
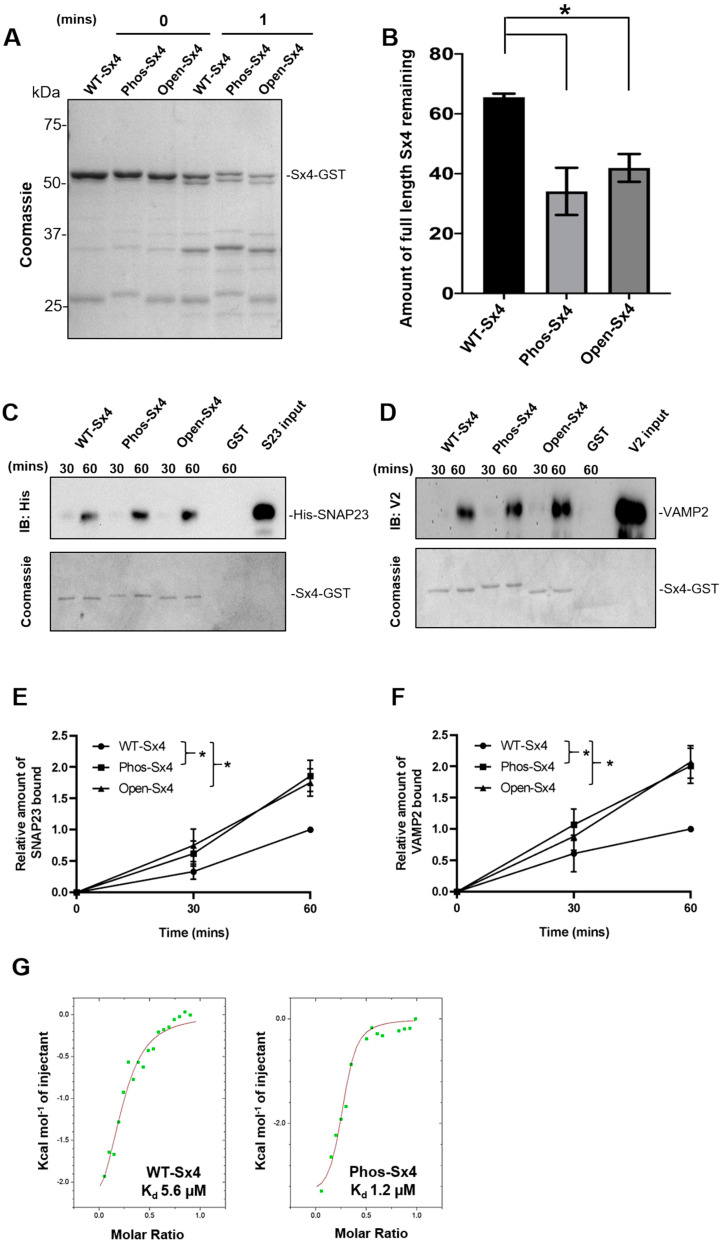
Phosphomimetic mutation of Sx4 alters the protein’s conformation and increases its binary interactions with SNAP23 and VAMP2 in vitro. (**A**) Limited proteolysis: 5 µM of wildtype Sx4-GST (WT-Sx4) or a mutant version thereof was incubated with 1.2 nM chymotrypsin at room temperature for the indicated times. SDS-PAGE followed by coomassie staining was used to visualise the levels of protein degradation. (**B**) Image J software was used to compare the remaining amounts of SX4-GST variants upon proteolysis (mean values expressed as a percentage from 3 experiments). In vitro binding assays: 1–5 μg of Sx4-GST, mutant versions thereof, or GST alone, immobilised on glutathione sepharose beads, were incubated with an excess of either His-SNAP23 or VAMP2 at 4 °C for the indicated times (min). Beads were harvested, and bound material analysed by immunoblotting to assess the ability of the Sx4–GST variants to form binary complexes with either His-SNAP23 (**C**) or VAMP2 (**D**). Coomassie staining was used to visualise the input proteins for each assay. ImageJ software was used to compare the amounts of His SNAP23 (**E**) and VAMP2 (**F**) to form binary complexes to different variants of Sx4-GST (normalised to the amounts of Sx4-GST-3 experiments) and plotted as a function of time. Error bars represent  ±  SEM, * *p* < 0.05. (**G**) WT-Sx4 (298 μM) or Phos-Sx4 (308.25 μM) was titrated with SNAP23 (64 μM) and the raw data from a typical ITC experiment is shown, using MicroCal VP-ITC instrument (Malvern). The K_d_ of the interaction was calculated by fitting the data with a nonlinear least squares routine using Microcal Origin software. Original Western Blotting Figures can be found in Appendix A.

**Figure 4 biomolecules-13-01738-f004:**
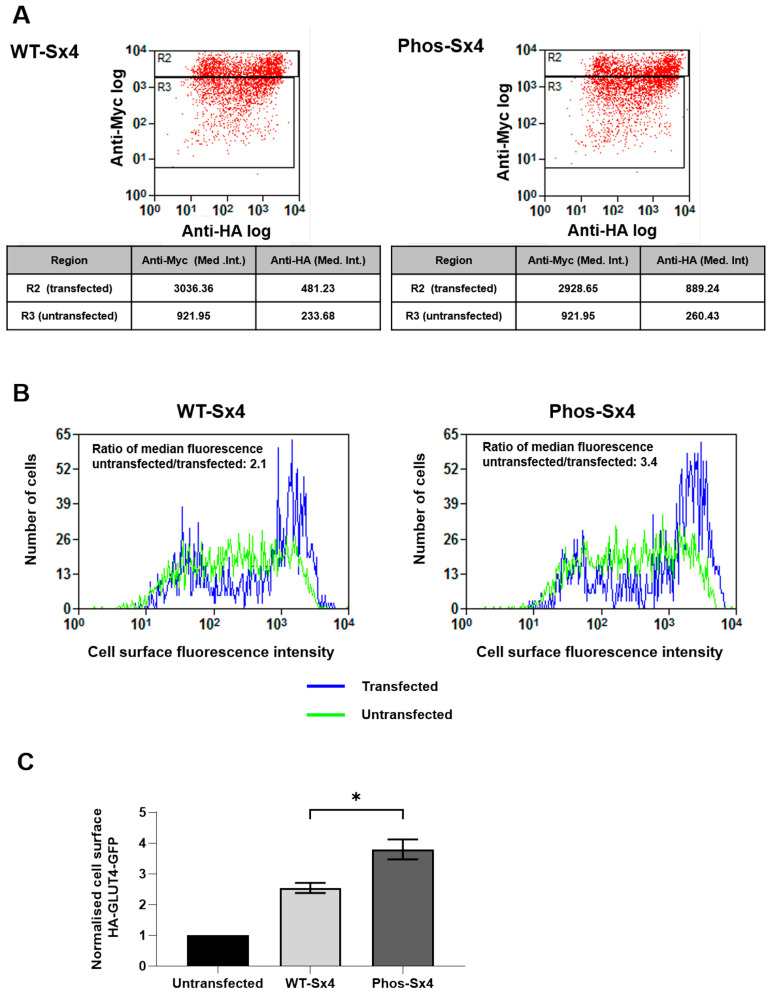
Phosphomimetic mutation of Sx4 increases cell surface GLUT4 in HeLa cells. HeLa cells stably expressing HA–GLUT4–GFP transiently transfected with an expression vector encoding either full-length myc-tagged wild-type Sx4 (WT-Sx4) or a phosphomimetic version with Y115 and Y251 mutated to glutamic acid residues (Phos-Sx4). HA–GLUT4–GFP at the cell surface was identified by staining for the HA-tag in the absence of cell permeabilization. Following subsequent re-fixing, permeabilisation, and staining for myc-tag present on the Sx4 constructs, flow cytometry was performed to quantify cell surface GLUT4 in both transfected and non-transfected cells. (**A**) scatter plots from a representative experiment. Gating distinguishing transfected (R2) and untransfected (R3) cell populations is shown along with median intensities of myc and HA staining (tables). (**B**) Shows histograms from a representative experiment of the fluorescence intensity (HA epitope) comparing untransfected (green) and transfected (blue) cells (ratio of the median fluorescence of cell surface HA-staining of untransfected/transfected cells is shown). (**C**) Quantification of cell surface GLUT4 in cells expressing myc-tagged versions of Sx4. A minimum of 10,000 cells were analysed for each experimental condition, reported as mean ± SEM, from 3 experiments, * *p* < 0.05.

**Figure 5 biomolecules-13-01738-f005:**
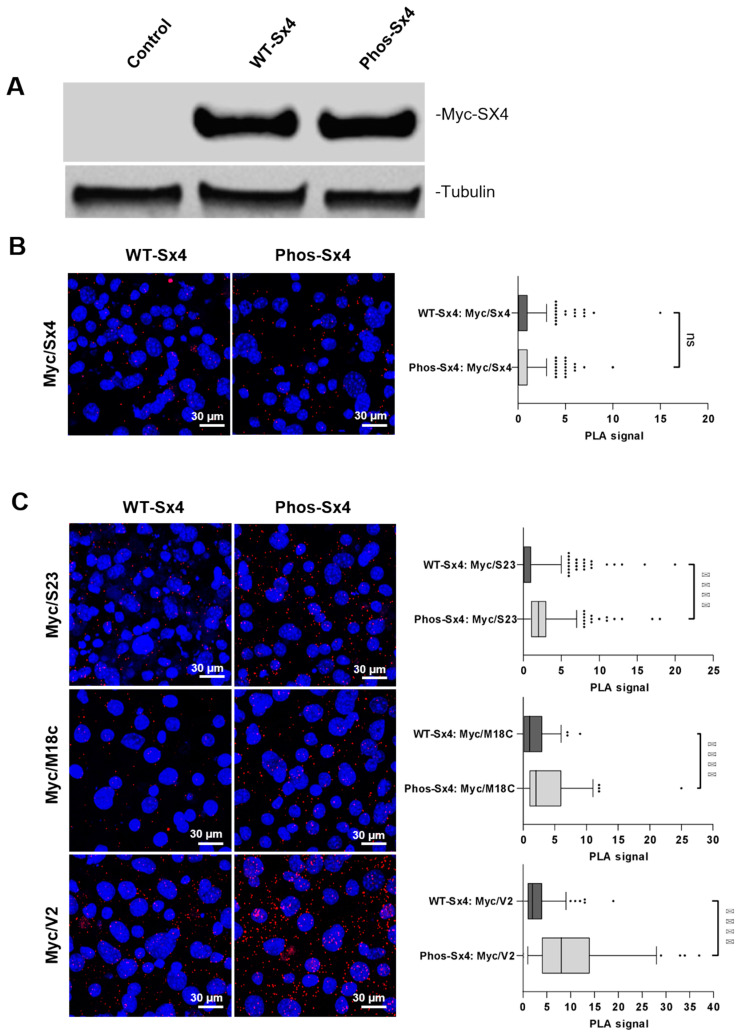
Phosphomimetic mutation of Sx4 shows enhanced binary interactions with SNAP23, VAMP2, and Munc18c in vivo. (**A**) Extracts prepared from 3T3-L1 adipocytes stably expressing full-length myc-tagged wildtype Sx4 (WT-Sx4), a phosphomimetic mutant version (Phos-Sx4) along with cells not overexpressing Sx4 (control) were subject to immunoblot analysis with anti-myc antibodies to identify overexpressed Sx4 constructs (Myc-Sx4 top panel) and tubulin as a loading control. The same cells were subjected to Proximity ligation assay (PLA) to detect indirectly the levels of the overexpressed Sx4 constructs Myc/Sx4) (**B**), or the pairwise associations of the Myc-tag Sx4 constructs with SNAP23 (Myc/S23), VAMP2 (Myc/V2) and Munc18C (Myc/M18c) (**C**). Cells were examined by confocal microscopy for PLA signal detection (red, blue = DAPI [4′,6-diamidino-2-phenylindole]). Cell Profiler software was used for PLA signal quantification. Box plots represent the median numbers of signals per cell from 300 to 500 cells per experiment. Images are representative of the results of 3 independent experiments, **** *p* < 0.001, ns *p* > 0.05. ns = not significant. Original Western Blotting Figures can be found in Appendix A.

## Data Availability

Data is contained within the article or Appendix A, or is available from the communicating author upon request.

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
