# Peer review of "Phosphorylation of Syntaxin 4 by the Insulin Receptor Drives Exocytic SNARE Complex Formation to Deliver GLUT4 to the Cell Surface"

_biomolecules, 2023, doi:10.3390/biom13121738_

Round 1

Reviewer 1 Report

Comments and Suggestions for Authors

This manuscript presents data supporting a model in which phosphorylation of Sx4 by the insulin receptor promotes the opening of this syntaxin to a fusion-competent state. This then facilitates the formation of a SNARE complex with Vamp2 and Snap23 to enhance the fusion of GLUT4-containing vesicles with the plasma membrane.  The data appear to be solid, and the manuscript will be of interest to investigators studying SNARE regulation and GLUT4 trafficking.

The investigators make note of ITC data, but do not present this, and ideally it should be shown.  This would help bring clarity to the discussion. On line 406, the authors state that phospho-Sx4 binds SNAP23 and VAMP2 with greater affinity than the wildtype Sx4 protein (Fig. 3). However, Figs. 3EF show a difference is in the kinetics of binding, not necessarily in the affinity per se. If the phosphorylation causes opening of the syntaxin, then this would increase the affinity, so to some extent this is a minor point. The authors note (lines 284-287) that ITC was done and that the Kd for Sx4-SNAP23 was ~4-fold lower with the phospho-mutant. This is a true thermodynamic measurement and showing the data will help make the authors’ point.

The authors discuss the experimental difficulty associated with expressing phospho-resistant Sx4 mutants in 3T3-L1 cells. This reviewer agrees that although these data would be helpful, the experiments are not required for publication of the present work. Even so, the authors might be slightly more cautious about the role of this step in overall GLUT4 trafficking.  For example, the abstract states “Delivery of GLUT4 to the cell surface is mediated by formation of functional SNARE complexes…”, which suggests that this is the main insulin-regulated step.  The sentence might be better phrased, “is mediated in part by formation of”, or perhaps “requires the formation of functional SNARE complexes”.  This is a minor point.

How do the authors think that phosphorylation of Y115 and Y251 might function?  Is Y115 in the Habc domain, and is Y251 in the core SNARE domain? Does Y115E reduce affinity of the Habc for the core SNARE domain? Some speculation may be permissible in the discussion.

Author Response

Please see uploaded Word document

Reviewer 2 Report

Comments and Suggestions for Authors

This is a very well-performed study investigating how tyrosine phosphorylation of Syntaxin 4 might influence its interaction with other SNARE proteins (VAMP2, SNAP-23, and Munc18c). Experiments performed rigorously show that: i) Two tyrosine residues (Y115, Y251) in Syntaxin 4 can be phosphorylated by a recombinant activated cytosolic insulin receptor kinase (CIRK); ii) Tyrosine phosphorylation by CIRK only occurs on Y115 and Y251; iii) A phosphomimetic of Syntaxin 4 more readily forms a Syntaxin 4-VAMP2-SNAP-23 complex; iv) The phosphomimetic undergoes more rapid proteolysis consistent with it existing in an open conformation; and v) It increases GLUT4 plasma membrane content in Hela and 3T3-L1 cells. This is beautiful work and no major issues/concerns exist, only a couple very minor grammatical errors.

Minor

Line 273: ‘from a closed to an open conformation closed’ Should the second ‘closed’ be deleted.

Fig. 3B: Could better align the bar showing the statistical difference between bars 1 and 3 to be over bar 3.

Line 320: ‘significantly more significantly more’ Guessing the second ‘significantly more’ should be deleted.

Author Response

Please see uploaded Word document
